# Genome-Wide Analysis of Q-Type C2H2 ZFP Genes in Response to Biotic and Abiotic Stresses in Sugar Beet

**DOI:** 10.3390/biology12101309

**Published:** 2023-10-04

**Authors:** Menglin Li, Xuanyu Dong, Guozhang Long, Zongying Zhang, Chenggui Han, Ying Wang

**Affiliations:** Ministry of Agriculture and Rural Affairs Key Laboratory of Pest Monitoring and Green Management, College of Plant Protection, China Agricultural University, Beijing 100193, China; s20173192156@cau.edu.cn (M.L.); donguxuanyu@cau.edu.cn (X.D.); sy202231933265@cau.edu.cn (G.L.); zhangzongying@cau.edu.cn (Z.Z.); hanchenggui@cau.edu.cn (C.H.)

**Keywords:** sugar beet, C2H2-type zinc finger protein, phylogenetic analysis, abiotic stress, biotic stress

## Abstract

**Simple Summary:**

A plant’s C2H2-type zinc finger proteins (C2H2-ZFPs) play crucial roles in the process of plant growth and development, as well as various stress responses. The Q-type ZFP family, which contains a conserved “QALGGH”, has been reported in many plants. Sugar beet is an important crop for sugar production. Salt stress and viral infection significantly reduce both sugar yield and processing quality of sugar beet. So far, the genome-wide analysis of Q-type C2H2 ZPFs and their expression pattern in sugar beet have not been analyzed yet. This study analyzed 35 Q-type ZFPs in sugar beet and their expression patterns under salt stress and virus. These results will provide theoretical evidence for understanding the functions of Q-type ZFPs.

**Abstract:**

A plant’s Q-type C2H2-type ZFP plays key roles in plant growth and development and responses to biotic and abiotic stresses. Sugar beet (*Beta vulgaris* L.) is an important crop for sugar production. Salt stress and viral infection significantly reduce the root yield and sugar content of sugar beet. However, there is a lack of comprehensive genome-wide analyses of Q-type C2H2 ZFPs and their expression patterns in sugar beet under stress. In this study, 35 sugar beet Q-type C2H2 ZFPs (BvZFPs) containing at least one conserved “QALGGH” motif were identified via bioinformatics techniques using TBtools software. According to their evolutionary relationship, the BvZFPs were classified into five subclasses. Within each subclass, the physicochemical properties and motif compositions showed strong similarities. A Ka/Ks analysis indicated that the *BvZFPs* were conserved during evolution. Promoter cis-element analysis revealed that most *BvZFPs* are associated with elements related to phytohormone, biotic or abiotic stress, and plant development. The expression data showed that the *BvZFPs* in sugar beet are predominantly expressed in the root. In addition, *BvZFPs* are involved in the response to abiotic and biotic stresses, including salt stress and viral infection. Overall, these results will extend our understanding of the Q-type C2H2 gene family and provide valuable information for the biological breeding of sugar beet against abiotic and biotic stresses in the future.

## 1. Introduction

Transcription factors (TFs) function as crucial molecular regulators for gene expression in all organisms [1]. In plants, a number of TF families have been identified, including bZIP [2], WRKY [3], NAC [4], MYB [5], and zinc finger proteins (ZFPs) [6], which play important roles in many biological processes, such as growth, development, reproduction, and stress responses. Among these, ZFPs represent one of the largest TF families in plants [6]. ZFPs contain a variable number of zinc finger domains, each of which consists of cysteine (Cys) and histidine (His) residues combined with a zinc ion to form a three-dimensional finger-type structure. According to the number and arrangement of Cys and His residues, ZFPs have been classified into C2H2, C2HC, C2HC5, CCCH, C3HC4, C4, C4HC3, C6, and C8 [6].

C2H2-type ZFPs, also known as TFIIIA-type zinc fingers, are among the most extensively studied and abundant ZFPs in eukaryotes [7]. Recently, the in silico genome-wide identification and functional characterization of plant C2H2-type ZFPs were well analyzed in many species, including 321 members in *Glycine max* [8], 301 in *Brassica rapa* [9], 218 in *Medicago truncatula* [10], 204 in *Triticum aestivum* [11], 189 in *Oryza sativa* [12], 176 in *Arabidopsis thaliana* [13], 150 in *Zea mays* [14], 145 in *Sorghum bicolor* [15], 129 in *Cucumis sativus* [16], 109 in *Populus trichocarpa* [17], and 104 in *Solanum lycopersicum* [18]. These findings indicate that C2H2 ZFPs are ubiquitous in the plant kingdom, playing important regulatory roles in various biologic processes, including development and organogenesis, as well as responses to stresses and defense. The C2H2-type ZFPs in *A. thaliana* are categorized into three sets (A, B, and C), with each set further divided into several different subsets, such as C1, C2, and C3 [19]. To date, the plant C1 subset has been the most extensively investigated, and its members have been further classified into five subclasses (C1-1i to C1-5i) based on the number of zinc finger domains [20]. Moreover, the majority of plant C2H2 ZFPs contain a highly conserved QALGGH sequence within their zinc finger domain (CX_2–4_CX_3_FX_3_QALGGHX_3–5_H), identifying them as the plant-specific Q-type subfamily of C2H2 ZFP.

The first Q-type C2H2 ZFP was discovered in *petunia*, and a total of 21 Q-type ZFPs were identified based on their specific structures [21,22]. The conserved QALGGH motif was found to be essential for DNA binding activity [23]. In addition to the conserved zinc finger domain, the N-terminal of some plant C2H2 ZFPs contains a B-box and an L-box motif, while the C-terminal has an ethylene-responsive element binding-factor-associated amphiphilic repression (EAR) motif. The B-box region acts as a nuclear localization signal (NLS). The L-box motif, usually consisting of a core sequence of EXEXXAXCLXXL, is thought to relate to protein–protein interactions. The EAR motif, also known as the DLN-box, has been identified as playing a role in transcriptional repression [24].

The plant Q-type C2H2 ZFP is involved in plant development and various abiotic stress responses, such as drought, salt, osmotic, low temperature, and oxidative stresses [25]. For example, the expression of *AZF1*, *AZF2*, *AZF3*, *STZ/ZAT10*, *ZAT11*, and *ZAT18* is strongly induced by drought, salt, cold stresses, or abscisic acid treatment [26], while the expression levels of *ZAT7* and *ZAT12* are upregulated by oxidative stress, heat shock, or wounding in *A. thaliana* [27]. *ZAT7* has been shown to be involved in plant growth suppression and increased tolerance to salinity stress depending on its EAR motif [28]. Overexpressed *Zat7* was more tolerant to salinity stress than seedlings of wild-type plants in *Arabidopsis* [28]. Transgenic plants overexpressing *Zat12* could tolerate oxidative stress in *Arabidopsis* [27]. More interestingly, *ZAT18* was initially identified as a positive regulator of drought stress tolerance in *Arabidopsis* [29]. A recent study indicated that *Pseudomonas syringae* induces *ZAT18* expression to repress the transcription of *EDS1* for bacterial infection [30]. *ZAT18* overexpressing plants were more susceptible to *Pst* DC3000 compared to Col-0, while *ZAT18-KO* plants displayed enhanced resistance in *Arabidopsis* [30]. In *T. aestivum*, 47 Q-type C2H2-ZFPs were identified, and the expression of the majority of *TaZFP* genes was responsive to drought stress in either leaf or root [31]. In *O. sativa*, *OsZFP179* was characterized as a salt-responsive gene, and it was found to enhance salt tolerance in transgenic rice plants [32]. Until now, more Q-type C2H2 ZFPs have been identified in *A. thaliana*, *T. aestivum*, *Brassica oleracea*, *S. lycopersicum*, and *Medicago. sativa* via genome-wide identification [8,20,31,33,34].

Sugar beet (*Beta vulgaris*), a member of the Amaranthaceae family [35], is an important crop in temperate climates zone, accounting for 20–30% of the world’s sugar production. Sugar beet also provides essential raw materials for bioethanol, animal feed, pulp, pectin extract, and functional-food-related industries [36]. Sugar beet is frequently subjected to various biotic and abiotic challenges that reduce both sugar yield and processing quality [37]. Salinity is a major abiotic stress that limits plant growth and development [38]. Although sugar beet is a salt-tolerant crop, prolonged exposure to salt stress can result in a significant yield loss in beet production [39]. In addition, rhizomania caused by beet necrotic yellow vein virus (BNYVV) stands as one of the most severe biotic threats to sustainable beet production globally [40]. Susceptible varieties infected by BNYVV exhibit the pronounced lateral rootlet proliferation of taproot and yellow veins on systemically infected leaves [41]. This leads to significant losses in root yield as well as a decline in sugar content. Given these challenges, the identification and analysis of sugar beet genes involved in abiotic and biotic stress is crucial, as it offers genetic resources for molecular breeding. Recently, the genome of industrial diploid sugar beet (2n = 18 chromosomes) was sequenced, accelerating sugar beet breeding for tolerances and resistances against abiotic or biotic stresses [35]. So far, the structure and function of the *BvbZIP* and *BvWRKY* family, as well as their expression pattern under salt stress, have been genome-wide analyzed for sugar beet [42,43]. However, information about Q-type C2H2-ZFPs in sugar beet remains unknown. Since Q-type ZFP genes play vital roles in plant development and stress responses, it is essential to identify and analyze the Q-type ZFP gene family in the sugar beet genome.

In this study, we identified 35 Q-type *ZFP* genes (*BvZFPs*) in sugar beet comprising different numbers of zinc finger domains. The phylogenetic relationships, genomic location, gene structure, chromosome distribution, gene duplication, and cis-regulatory elements are also explored. In addition, we determined their mRNA expression profiles in both leaf and root tissues under salt stress and BNYVV infection. This study aims to provide a comprehensive understanding of the Q-type C2H2 gene family and shed light on the roles of the *BvZFP* family in sugar beet, especially the functional characterization of plant development processes and stress responses. This research study can help improve plant quality, stress resistance, and enhanced crop production via genetic modifications in Q-type C2H2 gene family members in sugar beet.

## 2. Materials and Methods

### 2.1. Acquisition of the Sequences of the Q-Type ZFP Transcription Factor Family in Sugar Beet

The protein, cDNA, and genome annotation files of sugar beet were retrieved from the NCBI database (sugar beet EL10_1.0) [44]. The HMM (hidden Markov model) profiles of the C2H2 zinc finger domain (PF13912, PF12756, and PF00096) were extracted from the Pfam database (http://pfam.xfam.org/ (accessed on 15 March 2023)). Subsequently, C2H2 zinc finger protein candidates were identified based on the HMM profiles using Simple HMM Search in TBtools [45]. To further identify Q-type ZFPs, each full-length sequence was individually searched, and 35 Q-type sequences containing the QALGGH motif were retained as subjects for further investigation. Then, the identified ZFP sequences were further confirmed using SMART tool (http://smart.embl.de/smart/set_mode.cgi?NORMAL=1 (accessed on 22 March 2023)), InterPro (https://www.ebi.ac.uk/interpro/ (accessed on 22 March 2023)), and NCBI-CDD search (https://www.ncbi.nlm.nih.gov/cdd/ (accessed on 22 March 2023)). The physicochemical properties of the family members were analyzed using the online tool ExPASy Proteomics Server (https://web.expasy.org/protparam/ (accessed on 27 March 2023)). The subcellular localization analysis of each gene family member was carried out using the Plant-mPLoc (http://www.csbio.sjtu.edu.cn/bioinf/plant-multi/ (accessed on 27 March 2023)) online tool.

### 2.2. Chromosomal Distribution, Protein Characterization, and Amino Acid Properties

The starting and ending positions of all *BvZFP* genes on each chromosome were retrieved from the sugar beet gene annotation database, and the results were visualized using the Gene Location Visualize program of Tbtools. Out of 35 genes, 34 *BvZFP* genes were located on the nine chromosomes, while the remaining 1 may be located on unannotated intergenic regions of the genome. The members of sugar beet *BvZFP* were renamed according to the location on the chromosome. Protein properties, including the length of the amino acid (aa), molecular masses (MW), theoretical isoelectric point (pI), instability index, and subcellular localization, were predicted using ExPASy Server (https://web.expasy.org/protparam/ (accessed on 27 March 2023)).

### 2.3. Multiple Sequence Alignment and Phylogenetic Tree Construction

The full-length sequences of Arabidopsis C2H2 C1-2i zinc finger proteins (Q-type ZFPs) were downloaded from the TAIR database (http://www.arabidopsis.org/ (accessed on 16 March 2023)). The multiple sequence alignment of Q-type zinc finger protein members in sugar beet and Arabidopsis was performed using MEGA software with the default parameters. Full-length amino acid sequences were aligned using ClustalX, and the phylogenetic tree was constructed using the neighbor-joining method with the following parameters: bootstrap method with 1000 replicates and partial deletion.

### 2.4. Analysis of Members of the C2H2 ZFP Gene Family

The full-length protein sequences of each subfamily were analyzed for conserved motifs using Multiple Expectation Maximization for Motif Elicitation (MEME, https://meme-suite.org/meme/ (accessed on 22 March 2023)) [46]. We used the classic discovery mode and adjusted the parameters as follows: distribution of motifs = any number of repetitions, the maximum number of motifs = 10, and the optimum motif width range from 6 to 50 (inclusive). The gene structure and conserved motifs of C2H2 genes were visualized using Gene Structure View tools in Tbtools software.

### 2.5. Promoter Analysis of the BvZFP Genes in Sugar Beet

The 2000 bp promoter sequences upstream of the 35 *BvZFP* start codons were extracted from the sugar beet EL10_1.0 genome. The predicted cis-acting elements on promoters with their positional information were identified using the PlantCARE online tool (http://bioinformatics.psb.ugent.be/webtools/plantcare/html/ (accessed on 23 March 2023)). The obtained results were used to predict putative stress- and hormone-responsive cis-acting elements and visualized by Simple BioSequence Viewer in TBtools.

### 2.6. Collinearity Analysis of Arabidopsis and Sugar Beet BvZFP Genes

*Arabidopsis* was selected for collinearity analysis with sugar beet. The genome sequences and annotation files of Arabidopsis were downloaded from NCBI databases, and the chromosome length and location information for the ZFP genes on the genome of the two species were extracted. Using the One Step MCScanX tool of TBtools, we investigated gene replication events and collinearity relationships for gene pairs between two species. All data were visualized via the use of the Advanced Circos program of TBtools software. Ka and Ks substitution between gene pairs were also calculated using the Simple Ka/Ks Calculator tool.

### 2.7. BvZFP Genes Expression Network Analysis

The transcriptome data of sugar beet under salt stress and BNYVV were downloaded from the NCBI SRA database (accession number: PRJNA666117, https://www.ncbi.nlm.nih.gov/bioproject/PRJNA666117/ (accessed on 27 March 2023)), which could be used to analyze the expression pattern of *BvZFP* genes under 300 mM salt stress. In addition, the database of E-MTAB-8187 from NCBI was used to analyze the expression pattern of *BvZFP* genes under BNYVV infection. GraphPad Prism was used to map gene expressions.

To identify *BvZFP* genes between the two groups, the expression level of each transcript was calculated based on the number of fragments per kilobase of exons per million mapped reads (FPKM). RSEM (http://deweylab.biostat.wisc.edu/rsem/ (accessed on 1 April 2023) was used to determine gene abundance. The R statistical package software edgeR (empirical analysis of digital gene expression in R, http://www.bioconductor.org/packages/2.12/bioc/html/edgeR.html (accessed on 1 April 2023)) was used for differential expression analysis.

## 3. Results

### 3.1. Identification and Chromosomal Localization of the C2H2 Q-Type ZFP Subclass in Sugar Beet 

To identify C2H2 ZFP in the *B. vulgaris* genome, we utilized hidden Markov model (HMM) files (PF13912, PF12756, and PF00096) to conduct a genome-wide HMM search. This search yielded 104 putative non-redundant sugar beet C2H2 ZFP proteins (BvZFPs). This number was greater than that present in PlantTFDB, where 64 *BvZFP*s have been deposited for sugar beet (http://planttfdb.gao-lab.org/family.php?sp=Bvu&fam=C2H2 (accessed on 17 March 2023)). Subsequently, 35 Q-type ZFPs, characterized by the specific CX_2_CX_3_FX_3_QALGGHX_3–5_HX domain, were manually selected (Appendix A). According to the sugar beet genome database [47], we generated a map that detailed the physical positions of the Q-type *BvZFPs* (Figure 1). These Q-type *BvZFPs* were then renamed from *BvZFP1* to *BvZFP35* based on their physical positions on sugar beet chromosomes (Chr). All *BvZFP* genes were distributed widely and unevenly on nine Chrs, except for *BvZFP35*, which was located on unmapped scaffolds (Figure 1). Chr6 contained the largest number of *BvZFP* members and contained 8 *BvZFPs*, followed by Chr2, Chr3, Chr9, Chr1, Chr5, and Chr7, each of which contained 3 to 5 *BvZFPs*. Chr4 and Chr8 contained relatively fewer *BvZFP* members, with only two and one gene, respectively.

### 3.2. Characterization of the Sugar Beet C2H2 Q-Type ZFP Subclass

Based on the number of zinc finger domains and the spacing between the two His residues, the 35 *BvZFP* genes were divided into four groups, including 22 members in the C1-1i group (one zinc finger domain), 6 members in the C1-2i group, 6 members in the C1-3i group, and 1 member in the C1-4i group (Appendix A). Interestingly, several zinc finger domains with certain modifications of the “QALGGH” motif were observed for all 3i and 4i members (Appendix A). According to a previous study [16], the modified zinc finger domains were classified as the M-type. Furthermore, the protein properties of these genes were predicted using the ExPASy Server, and the results are shown in Table 1 and Appendix A. The protein lengths of the 1i group, 2i group, and 3i group ranged from 162 to 323, 186 to 456, and 237 to 572 amino acids for 1i group, 2i group, and 3i group, respectively. The molecular weights of the 1i group’s proteins ranged from 18,233 to 35,834 Da, with an average weight of 25,264 Da. For the 2i group, the molecular protein weights ranged from 20,667 to 50,063 Da, with an average weight of 32,905 Da. The average molecular weight of 3i group proteins is 50,602 Da, with individual weights ranging from 25,331 to 62,896 Da. For all BvZFPs, the theoretical isoelectric point (pI) fell between 5.69 and 9.22, and the instability index varied from 37.2 to 75.14. The GRAVY values, ranging from −1.27 to −0.358, revealed that sugar beet Q-type BvZFPs were hydrophilic proteins. In addition, subcellular localization prediction suggested that all BvZFPs are located in the nucleus.

### 3.3. Phylogenetic Analysis of Q-Type BvZFP Genes

To explore the evolutionary relationship between C2H2 Q-type genes in sugar beet and *A. thaliana* (*AtZFP*), we constructed a phylogenetic tree using MEGA10 based on the alignment of 93 Q-type ZFPs amino acid sequences at the whole protein level, which included 35 from sugar beet and 58 from *Arabidopsis*. The resulting tree classified the members into six major clades, including C1-1Q-A, C1-1Q-B, C1-2Q, C1-QM-A, C1-QM-B, and C1-QM-C (Figure 2). Remarkably, ZFP proteins that possessed the same types and numbers of zinc finger domains were clustered into the same clade. In the 1i group, 15 BvZFPs together with 16 AtZFPs were grouped in the C1-1Q-A clade, while 7 BvZFPs with 11 AtZFPs were grouped into C1-1Q-B clades. Both C1-1QA and C1-1QB clades contained one conserved “QALGGH” motif in their protein sequences. For the members of the 2i group, five BvZFPs along with 11 AtZFPs belonged to the C1-2Q clade. Following the same nomenclature as the C1-1Q clade, the members in the C1-2Q clade contained two conserved “QALGGH” motifs. The rest of the BvZFP members, which presented a variety of combinations with different numbers of Q- and M-type zinc finger domains, were defined as the C1-QM clades. For 3i and 4i group members, four BvZFPs were clustered into C1-QM-A clade together with nine AtZFPs, while four BvZFPs and seven AtZFPs were grouped into the clade C1-QM-B (Figure 2).

Moreover, an unrooted phylogenetic tree comprising 35 BvZFPs was constructed. Based on sequence similarity and tree topology, these BvZFPs were further classified into three clades, C1-1Q, C1-2Q, and C1-QM, and five subclades, including C1-1Q-A, C1-1Q-B, C1-2Q, C1-QM-A, and C1-QM-B (Figure 3A), according to the classification in Figure 2.

### 3.4. Gene Structure and Conservative Motif Analysis of C2H2 Q-Type BvZFPs

We analyzed the number of exon and intron structures of all 35 Q-type *BvZFP* genes. As shown in Figure 3, 30 *BvZFPs* were intronless, 4 *BvZFPs* contained a single intron, and only 1 *BvZFP* had four introns. All members with introns except for *BvZFP17* belonged to the 1i group. To better understand the characteristic regions of BvZFP proteins, we used the MEME online website to identify predicted conserved motifs. A total of 10 conserved motifs were identified, with their details presented in Appendix A. Motifs 1 and 2 were recognized as Q-type zinc finger domains (Figure 3D,F), while motif 4 represented the M-type zinc finger domains (Appendix A). Motif 1 was widely distributed in all 35 BvZFP proteins (Figure 3C), and motif 2 was found in members of the C1-2Q, C1-QM-A, and C1-QM-B clades (Figure 3D). Motif 4 was specific to the C1-QM clades. Motif 3, known as the EAR motif (Figure 3E), had conserved amino acid signatures “LxLxL” or “DLNx(1-2)P”. Among these BvZFPs, 18 BvZFPs contained the “LxLxL” type of EAR motif, while 14 BvZFPs contained the “DLNx(1-2)P” type of EAR motif. In addition, BvZFPs EAR motifs were predominantly located at the C-terminus region, except for BvZFP21 and BvZFP14. Motif 5, termed the L-BOX motif, was characterized by a core sequence of EXEXXAXCLXXL (Figure 3G). All 14 members in the C1-2Q and C1-QM clades contained Motif 5. Notably, some motifs were clade-specific. For example, motifs 10 and 6 mainly existed in C1-2Q. Motif 9 appeared in members of C1-2Q, C1-QM, and C1-1Q-B. Motif 8 was unique to C1-1Q-A (Figure 3). In summary, it was found that the BvZFPs clustered in the same subclades shared a similar motif composition, suggesting functional similarities among these evolutionarily conserved BvZFPs within the same clades.

### 3.5. Genomic Collinearity Analysis of C2H2 Q-Type BvZFPs between Sugar Beet and Arabidopsis

To investigate the potential evolution processes of C2H2 Q-type *BvZFP* genes, we analyzed the synteny relationship of 34 *BvZFPs* using TBtools. Intraspecific collinearity analysis showed that three pairs of *BvZFPs* exhibited collinearity within sugar beet. The ratio of the nonsynonymous substitution rate to the synonymous substitution rate (Ka/Ks) is widely used for evaluating the selective pressure of duplication events [34]. Generally, Ka/Ks > 1 indicates positive selection, Ka/Ks = 1 indicates neutral selection, and Ka/Ks < 1 indicates purification [48]. The Ka/Ks value of *BvZAT7* and *BvZAT32* is 0.1419 (Figure 4, Appendix A). Moreover, the phylogenetic tree analysis showed that these two genes were on the same branch. The results indicated that these synteny genes have undergone purifying selection in their evolutionary history. To further explore the evolutionary relationship of C2H2 Q-type *BvZFP*s, interspecific synteny comparisons between *B. vulgaris* and *A. thaliana* were also made. In our study, we identified 24 pairs of *ZFP* genes in sugar beet and *Arabidopsis* (Figure 5). The Ka/Ks ratios of these gene pairs ranged from 0.1 to 0.4, with an average of 0.25 (Appendix A), suggesting that the C2H2 Q-type *ZFP* gene remained relatively conserved in different species.

### 3.6. Promoter Analysis of the Q-Type BVZFP Genes in Sugar Beet

Cis-acting regulatory elements play essential roles in modulating the plant response to biotic and abiotic stresses. Therefore, the 2000 bp promoter regions located upstream of the *BvZFP* genes were extracted from the sugar beet EL10_1.0 genome database [44]. These sequences were then analyzed for cis-acting elements using the PlantCARE website. The results showed that 385 cis-regulatory elements belonging to 26 categories were obtained (Appendix A). The basal promoter elements, such as TATA-box and CAAT-box, and unannotated elements were excluded from this count. Most cis-elements were related to hormonal response and stress signal responsiveness (Appendix A). Among them, 197 elements were found to be related to hormonal responses, including 63 abscisic-acid-responsive elements (ABRE), 62 MeJA-responsive elements (TGACG-motif), 30 salicylic-acid-responsive elements (TCA-element), 29 gibberellin-responsive elements (TATC-box), and 13 auxin-responsive elements (TGA-element). Meanwhile, 46 elements associated with plant development, 47 elements that respond to abiotic stress, and 22 elements related to biotic stress were identified. In addition, the abiotic-stress-responsive elements were related to drought and low-temperature responses. As shown in Figure 6, 22 *BvZFPs* contain more than 10 functional elements from different categories. We also found that *BvZFP8*, *BvZFP34*, *BvZFP24*, and *BvZFP17* had a higher number of cis-regulatory elements, while *BvZAT2* and *BvZAT26* contained fewer cis-elements (Figure 6). The presence of these cis-regulatory elements in the *BvZFP* promoter regions suggested their involvement in the regulation of various plant pathways.

### 3.7. Expression Profiles Analysis of Q-Type BvZFPs in Different Tissue

To investigate the expression patterns of Q-type *BvZFP* genes in different tissues, we analyzed sugar beet transcriptome data for leaf and root tissues from three-pair-euphylla-stage sugar beet, which were taken from the SRA database [49]. A heatmap was constructed to visualize the expression patterns via TBtools software (Figure 7, Appendix A). In the control group, some *BvZFPs* displayed a tissue-specific expression pattern. Specifically, *BvZFP3*, *BvZFP8*, *BvZFP9*, *BvZFP16*, *BvZFP18*, *BvZFP19*, *BvZFP32*, and *BvZFP35* were only expressed in roots. Meanwhile, *BvZFP2*, *BvZFP6*, *BvZFP7*, *BvZFP14*, *BvZFP17*, *BvZFP20*, *BvZFP27*, *BvZFP28*, *BvZFP30*, and *BvZFP34* were expressed in both leaf and root tissues, while 17 *BvZFPs* did not show expression in either tissues. Interestingly, the expression levels of all *BvZFPs* in root tissues were generally higher than that in leaf tissue, except for *BvZFP14*.

### 3.8. Responses of Q-Type BvZFP Genes under Salt Treatment and Viral Infection

To further provide insight into the response of *BvZFPs* to abiotic and biotic stress, transcriptome data from the SRA database [49] were used to investigate the expression patterns of *BvZFPs* under salt stress or viral infection (Appendix A). Under a 300 mM NaCl treatment, the expression of most *BvZFPs* was changed. At 72 h after treatment, *BvZFP2*, *BvZFP6*, *BvZFP17*, *BvZFP30*, and *BvZFP34* were significantly up-regulated in leaves, while *BvZFP2*, *BvZFP14*, *BvZFP16*, and *BvZFP34* were significantly up-regulated in roots (Figure 7). Comparing the expression patterns between root and leaf tissues, we found that *BvZFP2* and *BvZFP34* were up-regulated in both tissues, suggesting their important role in response to salt stress in these tissues. 

Transcription factors in plants also play a pivotal role in the response to pathogen infections. Using beet necrotic yellow vein virus-infected sugar beet transcriptome data from the database [50], It was revealed that nearly half of the *BvZFP* genes exhibited differential expression at varying degrees. Among these genes, *BvZFP6*, *BvZFP8*, *BvZFP28*, and *BvZFP34* were significantly up-regulated, while *BvZFP2*, *BvZFP16*, *BvZFP17*, and *BvZFP30* were moderately induced. Conversely, the expression levels of *BvZFP13*, *BvZFP25*, *BvZFP26*, and *BvZFP33* were down-regulated (Figure 8). The above results indicated that certain *BvZFP* genes may be involved in either promoting or inhibiting viral infection by regulating downstream target genes.

## 4. Discussion

C2H2 ZFPs are one of the most extensively studied transcription factors that play crucial roles in many biological processes in eukaryotic organisms [6,21]. The Q-type ZFP, a plant-specific subfamily of C2H2-ZFPs, was found to be involved in plant development, as well as various stress responses [15,34,51]. So far, Q-type ZFPs have been extensively studied in many species, such as *A. thaliana*, *S. lycopersicum*, *Solanum tuberosum*, *T. aestivum*, *O. sativa*, and *P. trichocarpa* (Table 2). However, this subfamily has yet to be explored in *B. vulgaris*, one of the most important crops for sugar production. In this study, we performed a comprehensive genome-wide investigation of Q-type C2H2 *ZFPs* in sugar beet.

A total of 104 C2H2 *BvZFPs* were first identified from the current *B. vulgaris* genome using bioinformatic analysis. This number of *BvZFP* was lower than most previously studied species, except for *Vitis vinifera* and *C. annuum* (Table 2). Among these genes, 35 Q-type C2H2 *BvZFPs*, with each containing at least one conserved “QALGGH”-type zinc finger domain, were selected for further investigation. As expected, sugar beet also has the fewest Q-type *ZFP* genes compared with other sequenced plant genomes (Table 2). These Q-type *ZFP* genes were renamed from *BvZFP1* to *BvZFP35* based on their physical positions on the sugar beet chromosomes. Thirty-four members were unevenly distributed in all nine chromosomes, except for *BvZFP35* (LOC104893654) (Figure 1). Based on the number of zinc finger domains and the spacing between the two His residues, 63% (22 out of 35) members belonged to C1-1i group, 34% of members existed in both C1-2i and C1-3i groups, and only one gene *BvZFP22* was categorized in the C1-4i group. We did not identify any Q-type ZFPs in sugar beet that possessed five zinc finger domains, which are present in *Arabidopsis* and rice genomes [12,20]. Interestingly, we found several zinc finger domains with certain modifications to the “QALGGH” motif (M-type) for all 3i and 4i members. The lengths of the BvZFP proteins differed among groups, ranging from 162 to 323, 186 to 456, and 237 to 572 amino acids for 1i group, 2i group, and 3i group, respectively. All Q-type BvZFPs were predicted to be localized within the nucleus, indicating that these proteins indeed function as transcription factors in the nucleus. Phylogenetic analyses of 35 BvZFPs, using 58 *Arabidopsis* ZFPs as templates, classified 35 Q-type *BvZFP* genes into five major clades according to the number of “KS/KA/RS/RA/QA-LGGH” motifs they contained, including C1-1Q-A, C1-1Q-B, C1-2Q, C1-QM-A, and C1-QM-B (Figure 2). Proteins clustered into the same clade may have similar functions under various stresses and close evolutionary relationships. Among them, C1-2Q clusters with *Arabidopsis* C1-2i, which is highly conserved during evolution. It has been reported that mutations in the QALGGH sequence greatly affect DNA-binding activity [59]. Cis-acting elements are involved in the regulation of gene activity and serve as fundamental molecular switches during transcriptional regulation [60]. A previous study confirmed that many of these elements, such as ABREs and DREs, have been reported to widely participate in abiotic stress responses in *Artemisia annua*, *A. thaliana*, and *T. aestivum* [31]. In plants, hormones, such as auxin, abscisic acid, and gibberellin, play an important role in the growth and development of plants and the response to adversity stress [61,62]. However, no study has investigated these key regulatory elements in sugar beet. Among the *BvZFP* gene family members, 385 cis-regulatory elements were identified, which contain elements related to hormone response (abscisic acid, gibberellin, ethylene, and auxin), stress response elements (low temperature and drought), and growth- and development-related response elements. These results spotlight potential candidate genes for anti-abiotic stress, although their specific functions need to be confirmed by further investigation. Previous studies have shown that genes with fewer introns are more prone to be activated in response to stress [63]. In this study, approximately 86% (30/35) of *BvZFPs* have no introns, except for five *BvZFPs*, which indicates that *BvZFPs* could rapidly react to external stimuli. Collinearity analysis, an essential analytical strategy in comparative genomics, illuminates both large-scale and small-scale molecular evolutionary events across species. In this study, about 70% of the 34 mapped *BvZFP* genes are collinear with respect to four C2H2-ZFP genes on *Arabidopsis* chromosomes. The Ka/Ks ratios showed the conservation of BvZFPs throughout evolution, aligning with the previous views that regarded the C2H2 family as evolutionarily stable [64,65]. All this detailed information helps us better understand and screen for appropriate C2H2 gene family members in sugar beet.

Besides the conserved zinc finger domains, many previously identified plant ZFP proteins have EAR motifs. These motifs, characterized by conserved amino acid signatures “LxLxL” or “DLNxxP” in their respective C-terminal regions [66], play a key role in transcriptional repression [67,68]. It has been reported that 26% of wheat TaZFP members contained potential EAR motifs [31], while approximately 55% of grape VviZFP members possessed this motif [56]. Genome-wide analysis suggests that the EAR motif is conserved across all plant species and is involved in developmental and stress-related processes [69]. For instance, an EAR repressor named NIMIN1 negatively regulates the expression of the *PR1* defense gene in *Arabidopsis* [70]. *AtZP1*, which contains an EAR motif, negatively regulates *Arabidopsis* root hair initiation and elongation [71]. Most strikingly, our results reveal that 32 BvZFPs have at least one EAR motif, accounting for 91% of BvZFP members. As a key repression domain, alterations to any individual residue within the EAR can reduce or completely eliminate its transcriptional repression capability [72].

Previous studies demonstrated that many plant Q-type *ZFP* genes exhibited different expression patterns across various tissues [73,74,75,76]. For example, the Q-type genes of cucumber could be clustered into four groups according to their expression levels; *CsZFP* genes in group 1 showed high expression levels, while *CsZFPs* in group 4 exhibited little to no expression in all tissues [16]. In potatoes, some Q-type *StZFP* genes displayed a tissue-specific expression pattern. Specifically, *PG0005486* and *PG0030311* were predominantly expressed in leaves and roots, respectively [58]. In strawberries, *FaZAT10* was highly expressed in roots, followed by leaves and stems [74]. Moreover, RNA-seq data analysis for Q-type *TaZFPs* in different wheat tissues showed that 75% of *TaZFPs* were predominantly expressed in roots, implying their potential role in regulating wheat root development [31]. In the current study, we also found that many *BvZFPs* displayed a specific expression pattern between leaves and roots (Figure 6). With respect to the three-pair-euphylla-stage seedlings of sugar beet, 46% *BvZFPs* (16 out of 35) were not observed to be expressed in either leaf or root tissue, suggesting that these genes might not influence development at this growth stage. In contrast, 23% *BvZFPs* (8 out of 35 genes) were expressed preferentially in roots while 29% *BvZFPs* (10 out of 35) were expressed in both leaf and root tissues to variable degrees. Among the genes expressed in both tissues, root expression levels for *BvZFPs* were relatively higher than in leaf tissue except for one gene. In addition, *BvZFP6*, *BvZFP27*, and *BvZFP30*, each containing two “QALGGH” domains, were significantly expressed in roots. Therefore, the results suggested that many Q-type *BvZFP* genes are involved in sugar beet development under normal growth conditions, especially root development.

Plants are frequently affected by various abiotic and biotic. To adapt to these stresses, plants have evolved sophisticated mechanisms. Numerous Q-type ZFP genes have been found in response to diverse stresses, such as drought, salt, osmotic, low temperature, oxidative stress, and pathogen infection in many plant species [16,30,32]. Studies have shown that the overexpression of MdZAT5 promotes the expression of anthocyanin-biosynthesis-related genes *NHX1* and *ABI1* to actively regulate anthocyanin synthesis and increases sensitivity to salt stress in apple calli and *Arabidopsis* [75]. The ZFP gene *Csa6G303740* was significantly up-regulated in response to drought, heat, and salt stresses, indicating it could be involved in the regulation of abiotic stresses in cucumbers [16]. Potato *StZFP1*, which responds to salt, dehydration, and infection by *Phytophthora infestans*, has the potential to improve salt tolerance in transgenic tobacco [76]. 

In this work, we analyzed publicly available transcriptome data and found that five *BvZFP* genes, *BvZFP2*, *BvZFP6*, *BvZFP17*, *BvZFP30*, and *BvZFP34*, were up-regulated in leaves or/and roots under salt treatment, indicating that these genes may play roles in salt response in sugar beet. These genes belong to the C1-2i and C1-3i groups, and all contain two Q-type zinc finger domains. In addition, *BvZFP2*, *BvZFP17*, and *BvZFP* 30 are grouped into the same C1-QM clade, while *BvZFP6* and *BvZFP34* are clustered into the C1-2Q clade. For biotic stress, twelve *BvZFP* genes were differentially expressed by BNYVV infection. Interestingly, *BvZFP2*, *BvZFP6*, *BvZFP17*, *BvZFP30*, and *BvZFP34* were induced either by salt treatment or viral infection. On the other hand, some *BvZFP* genes only responded to viral infection; for example, *BvZFP8* and *BvZFP28* were significantly up-regulated in the BNYVV-infected samples, and *BvZFP13*, *BvZFP25*, *BvZFP26*, and *BvZFP33* were down-regulated exclusively due to viral infection. These four down-regulated genes all belong to the C1-1i group and are clustered into the C1-1Q-A clade. *BvZFP8* and *BvZFP28* are in the C1-1Q-B and C1-2Q clades, respectively.

Generally, members in the same cluster may have conserved functions. Previous studies have shown that *Arabidopsis ZAT6* is strongly induced by various stresses, including salt, cold, dehydration, and pathogen infection [77]. Plants overexpressing *AtZAT6* improved resistance to salt, drought, freezing stresses, and pathogen infection. In our study, the *BvZFP30* clustered with *AtZAT6* in C1-QM was differentially expressed under salt in sugar beet, indicating that BvZFP30 may have functions that are similar to AtZAT6 under abiotic stress. *ZAT18* was initially shown to be a positive regulator of drought stress tolerance in *Arabidopsis* [29]. A recent study has indicated that *Pseudomonas syringae* induces *ZAT18* expression to repress *EDS1* transcription during bacterial infection [30]. Furthermore, *BvZFP6*, *BvZFP34*, and *BvZFP28*, which are clustered with *ZAT18*, are induced by BNYVV infection. Given that both *BvZFPs* and *ZAT18* are highly expressed under the attack of pathogens, it is plausible that these genes may have similar functions with respect to responding to biotic stress. Taken together, these results indicate that *BvZFP* genes are involved in plant development and stress resistance in sugar beet, particularly in response to both abiotic and biotic stresses.

## 5. Conclusions

In summary, we performed a genome-wide exploration of Q-type ZFPs in sugar beet, and a total of 35 Q-type *BvZFP* genes that contain at least one conserved “QALGGH” motif were identified. We analyzed the physiochemical properties, genomic location, gene duplication event, gene structure, conserved motif compositions, and cis-regulatory elements of these *BvZFP* genes. Phylogenetic analysis revealed that 35 Q-type *BvZFP* genes were classified into five subclades. Moreover, the BvZFPs clustered within the same clade generally shared a similar motif composition. In addition, the expression profiles of *BvZFPs* in leaf and root tissues, as well as their responses to salt stress and viral infection, were analyzed. Q-type *BvZFPs* are predominantly expressed in roots and are enriched with members that are responsive to abiotic and biotic stresses. These results will enhance our understanding of the *BvZFP* gene family and provide valuable information for the further functional analysis of Q-type BvZFPs relative to the abiotic and biotic stress tolerance of sugar beet. Meanwhile, this study provides a theoretical basis for the biological breeding of sugar beet against salt stress and viral infection in the future.

## Figures and Tables

**Figure 1 biology-12-01309-f001:**
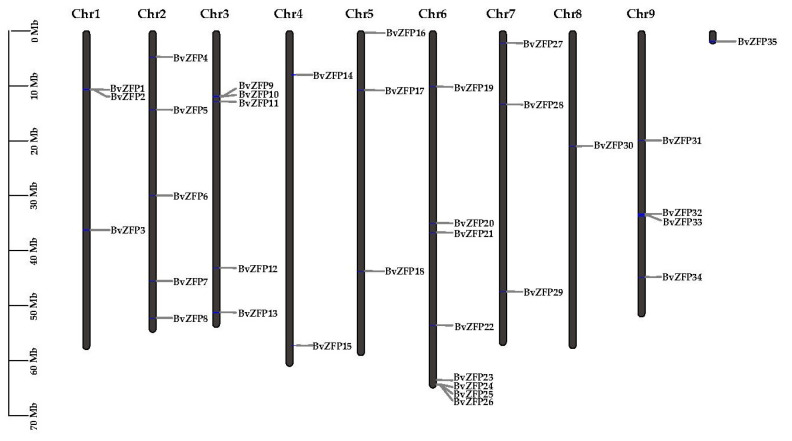
Chromosomal localization of Q-type *BvZFP* genes on sugar beet chromosomes. The 34 members are distributed over 9 chromosomes.

**Figure 2 biology-12-01309-f002:**
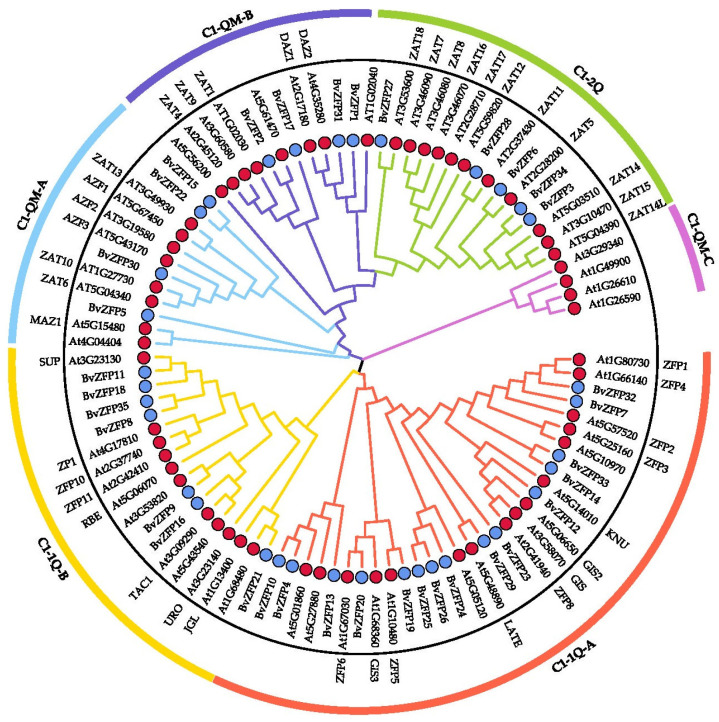
Phylogenetic relationships of Q-type C2H2-ZFPs between sugar beet and Arabidopsis. The unrooted phylogenetic tree was constructed using MEGA10 via the neighbor-joining method with 1000 bootstrap replicates. The tree was divided into six phylogenetic clusters. The red circles represent Arabidopsis ZFPs, the blue circles represent sugar beet ZFPs.

**Figure 3 biology-12-01309-f003:**
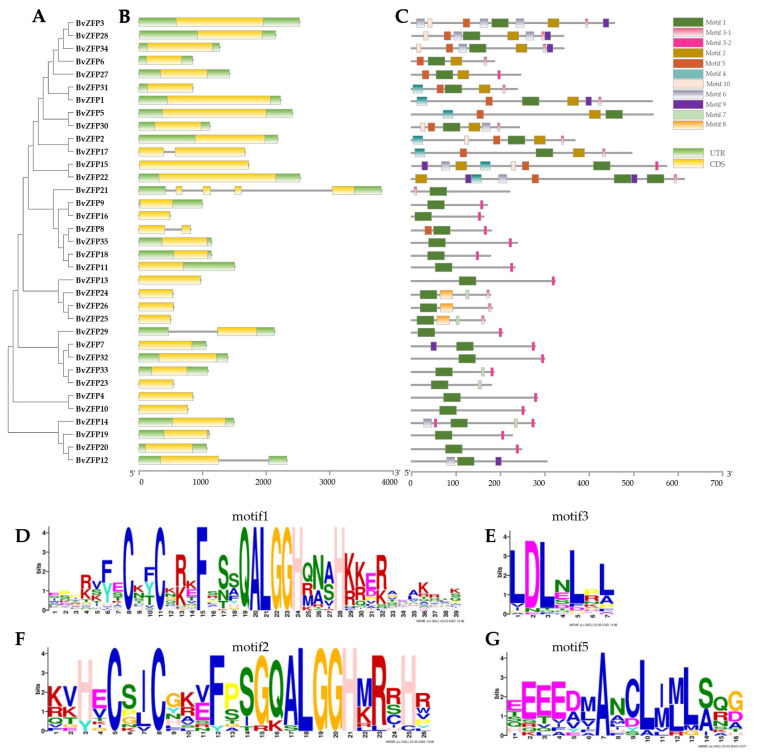
Motif distributions and gene-structure analysis of Q-type *BvZFP* genes. (**A**) The phylogenetic tree was built using the NJ method with a bootstrap value of 1000. (**B**) Exons, introns, and the untranslated region (UTR) are represented by yellow rectangles, gray lines, and green rectangles, respectively. (**C**) The conserved motifs in BvZFP proteins (1–10) are shown in different colors. The gray lines represent relative protein lengths. (**D**–**G**) The common conserved motifs in BvZFP proteins.

**Figure 4 biology-12-01309-f004:**
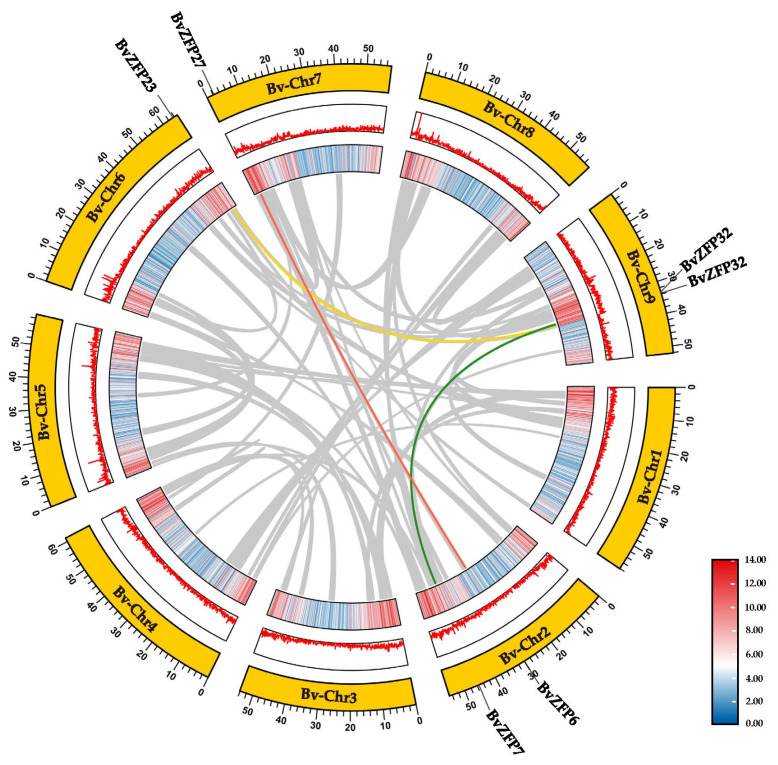
Distribution and collinearity of Q-type *BvZFP* genes in sugar beet. The outer circle represents the location of *BvZFP* genes on the chromosomes, and the inner circle histogram and heat map represent gene density. The grey lines represent all collinear genes on the sugar beet genome, and the colored lines connect the collinear *BvZFP* gene pairs.

**Figure 5 biology-12-01309-f005:**
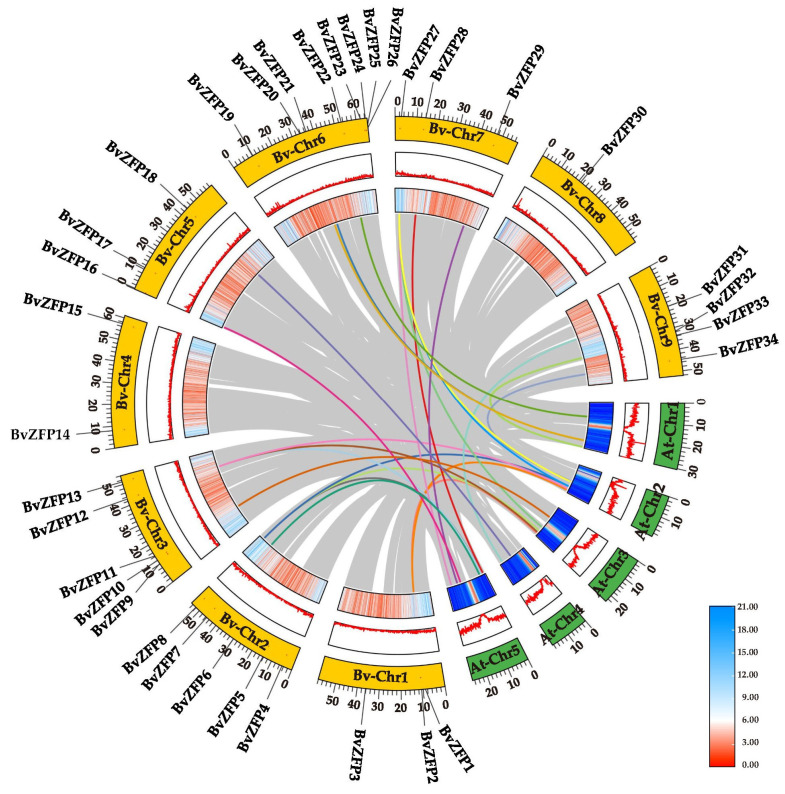
Collinearity analysis of Q-type C2H2-ZFPs in sugar beet and *Arabidopsis* genomes. The outer circle represents the different chromosomes of the two species, and the inner circle histogram and heat map represent the gene density of each species. The grey curves indicate the collinear gene regions within the genomes of the two species, and the colored curves emphasize the specific collinearity relationships among Q-type *BvZFP* genes between the two species.

**Figure 6 biology-12-01309-f006:**
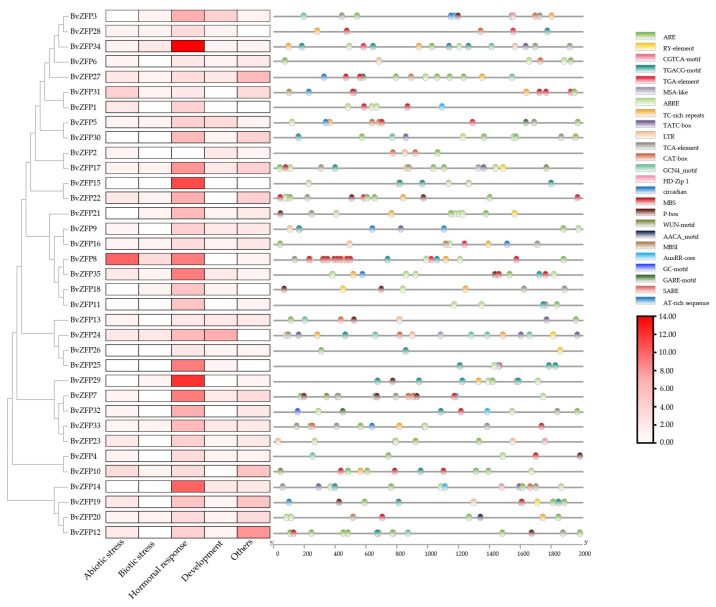
Distribution of cis-acting elements in the promoter of Q-type BvZFP members in sugar beet. The phylogenetic tree was developed on MEGA 10 using neighbor-joining phylogenetic method analysis. Both the bootstrap test and the approximate likelihood ratio test were set to 1000 times.

**Figure 7 biology-12-01309-f007:**
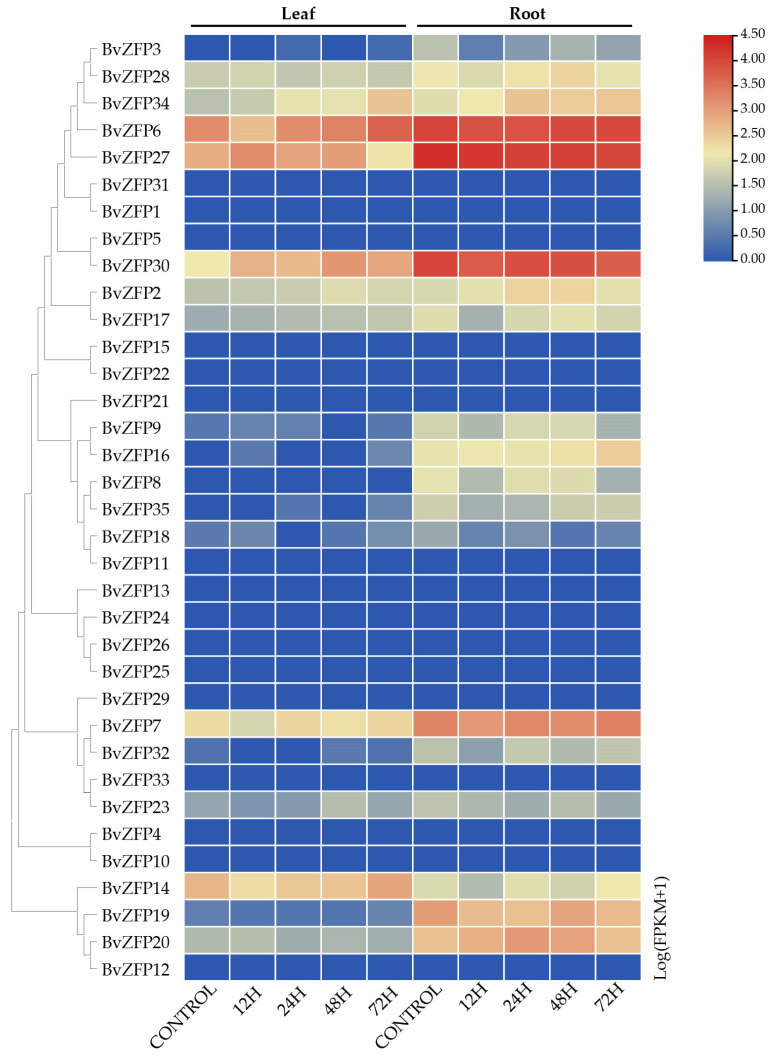
Transcript abundance of Q-type *BvZFPs* under salt stress. NaCl stress: 300 mM NaCl for 0, 12, 24, 48, and 72 h; numbers indicate three biological replicates. The heatmap represents the log2 (FPKM + 1) value.

**Figure 8 biology-12-01309-f008:**
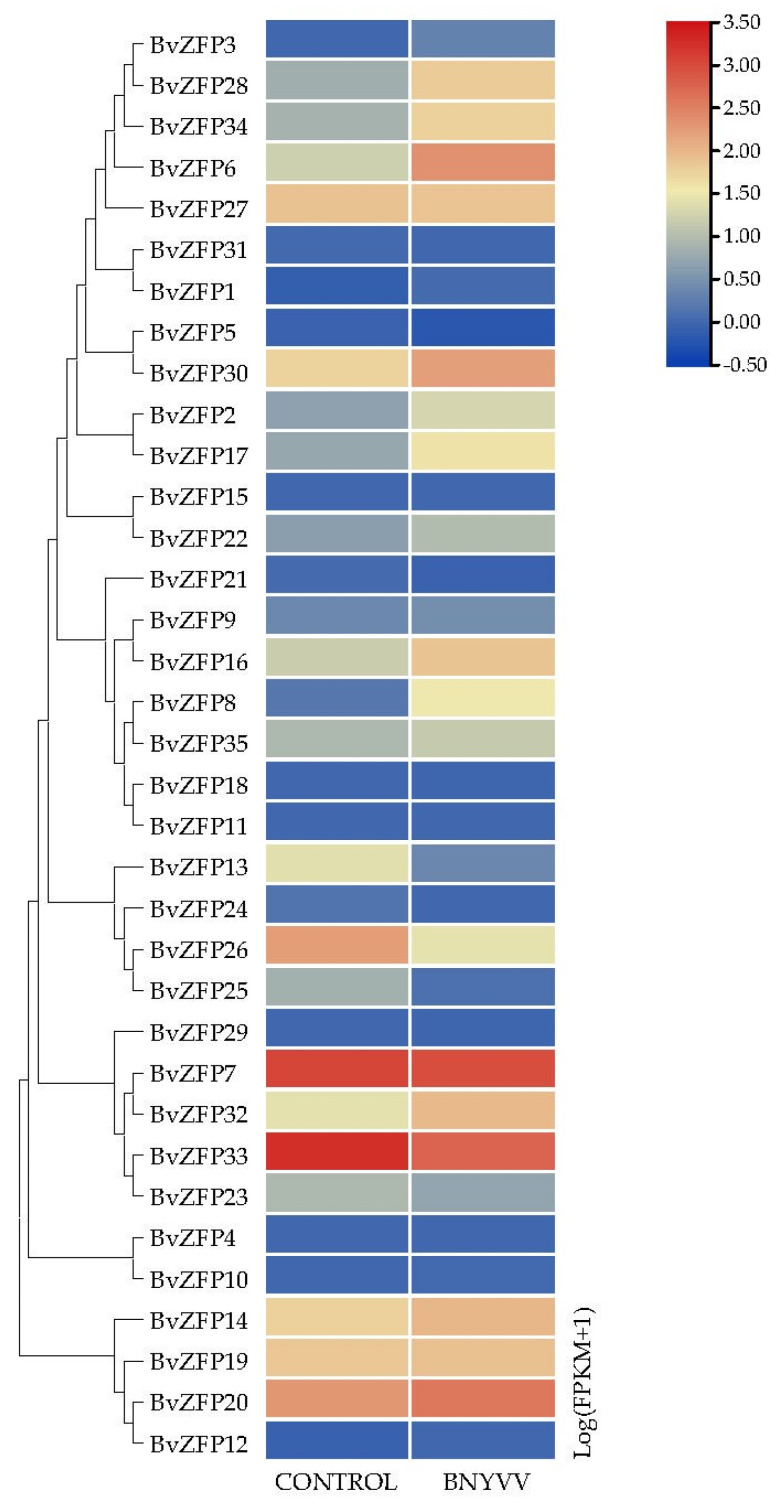
Transcript abundance of Q-type *BvZFPs* under BNYVV infection. Numbers indicate three biological replicates. The heatmap represents a log2 (FPKM + 1) value.

**Table 1 biology-12-01309-t001:** The physical and chemical properties of Q-type *BvZFPs* in sugar beet.

Gene Name	Gene ID	Amino Acid/aa	ORF/bp	Molecular Weight	pI	Instability Index	GRAVY	Location		Subcellular Localization	Group
*BvZFP1*	LOC104905560	541	1623	58,633.75	6.86	57.91	−0.786	10,695,460	10,697,687	Nucleus	3i
*BvZFP2*	LOC125495709	367	1101	41,279.25	8.02	55.83	−0.862	10,706,247	10,708,429	Nucleus	3i
*BvZFP3*	LOC104898669	456	1368	50,063.48	6.01	60.88	−1.002	36,219,959	36,222,487	Nucleus	2i
*BvZFP4*	LOC104887623	283	849	31,507.39	8.21	41.25	−0.667	4,757,460	4,758,311	Nucleus	1i
*BvZFP5*	LOC104906182	543	1629	60,544.44	7.17	59.45	−0.701	14,363,882	14,366,296	Nucleus	3i
*BvZFP6*	LOC104886969	186	558	20,667.45	8.86	57.6	−0.506	29,939,770	29,940,614	Nucleus	2i
*BvZFP7*	LOC104883124	278	834	30,386.53	6.13	59.97	−0.644	45,479,878	45,480,934	Nucleus	1i
*BvZFP8*	LOC104883624	179	537	20,615.07	8.3	54.73	−0.976	52,215,194	52,216,003	Nucleus	1i
*BvZFP9*	LOC104906611	170	510	18,959.88	6.65	65.03	−0.922	11,875,979	11,876,976	Nucleus	1i
*BvZFP10*	LOC104906657	255	765	28,493.69	9.22	57.02	−0.867	11,959,217	11,959,984	Nucleus	1i
*BvZFP11*	LOC104888927	232	696	26,701.97	6.89	67.24	−0.791	12,886,752	12,888,260	Nucleus	1i
*BvZFP12*	LOC104906753	304	912	33,684.31	8.85	43.09	−1.037	43,115,842	43,118,169	Nucleus	1i
*BvZFP13*	LOC104889553	323	969	35,834.41	8.56	58.9	−0.805	51,214,957	51,215,928	Nucleus	1i
*BvZFP14*	LOC104907233	277	831	31,391.42	6.2	58.48	−0.972	8,025,899	8,027,395	Nucleus	1i
*BvZFP15*	LOC104890281	572	1716	62,896.02	5.69	58.76	−1.132	57,253,629	57,255,347	Nucleus	3i
*BvZFP16*	LOC104894695	162	486	18,233.83	6.1	58.33	−1.089	405,412	405,900	Nucleus	1i
*BvZFP17*	LOC104893969	496	1488	54,927.45	8.48	66.19	−0.718	10,823,448	10,825,115	Nucleus	3i
*BvZFP18*	LOC104892924	177	531	20,328.6	5.97	61.13	−0.759	43,735,292	43,736,429	Nucleus	1i
*BvZFP19*	LOC104897252	226	678	25,693.54	9.1	57.43	−0.892	10,200,725	10,201,830	Nucleus	1i
*BvZFP20*	LOC104896450	247	741	25,939.68	8.64	46.76	−0.604	34,971,928	34,972,995	Nucleus	1i
*BvZFP21*	LOC104896549	220	660	24,314.02	7.24	54.26	−0.921	36,683,224	36,687,039	Nucleus	1i
*BvZFP22*	LOC104895813	613	1839	68,769.06	6.91	48.17	−1.27	53,568,797	53,571,334	Nucleus	4i
*BvZFP23*	LOC104895001	179	537	19,911.57	8.99	75.14	−0.496	63,541,505	63,542,044	Nucleus	1i
*BvZFP24*	LOC104885426	177	531	19,942.29	6.15	49.73	−0.751	64,317,253	64,317,786	Nucleus	1i
*BvZFP25*	LOC104885142	165	495	18,957.43	7.75	37.2	−0.788	64,323,220	64,323,717	Nucleus	1i
*BvZFP26*	LOC104884472	181	543	20,434.76	8.32	43.2	−0.842	64,406,993	64,407,538	Nucleus	1i
*BvZFP27*	LOC104900166	245	735	26,613.84	6.01	65.67	−0.521	2,250,266	2,251,688	Nucleus	2i
*BvZFP28*	LOC104908846	340	1020	37,135.81	7.75	62.32	−0.839	13,404,684	13,406,826	Nucleus	2i
*BvZFP29*	LOC104899451	204	612	22,977.76	8.79	47.89	−0.777	47,440,915	47,443,048	Nucleus	1i
*BvZFP30*	LOC104901462	242	726	25,661.67	8.09	62.93	−0.545	21,006,400	21,007,514	Nucleus	2i
*BvZFP31*	LOC104904162	237	711	25,331.76	8.55	54.27	−0.358	19,929,210	19,930,057	Nucleus	3i
*BvZFP32*	LOC104904895	299	897	33,574.34	8.24	50.67	−0.682	33,331,932	33,333,327	Nucleus	1i
*BvZFP33*	LOC104904924	185	555	21,039.05	8.76	59.64	−0.511	33,657,441	33,658,524	Nucleus	1i
*BvZFP34*	LOC104902980	342	1026	37,289.83	6.21	58.69	−0.823	44,803,778	44,805,047	Nucleus	2i
*BvZFP35*	LOC104893645	237	711	26,905.18	8.52	67.93	−0.792	1,955,234	1,956,376	Nucleus	1i

**Table 2 biology-12-01309-t002:** The number of C2H2 and Q-type zinc finger proteins in different species.

Species	The Number of C2H2 ZFPs	The Number of Q-Type	The Number of EAR Motif	Reference
*G. max*	321	130	--	[8]
*B. rapa*	301	110	--	[9]
*M. sativa*	--	58	--	[34]
*M. truncatula*	218	71	106	[10]
*Z. mays*	326	147	--	[52]
*T. aestivum*	204	159	68	[11]
*O. sativa*	189	72	43	[12]
*A. thaliana*	176	64	--	[20]
*S. bicolor*	145	--	--	[15]
*C. sativus*	129	52	--	[16]
*Dactylis glomerata*	125	50	--	[53]
*Setaria italica*	124	97	--	[54]
*Triticum turgidum*	122	96	11	[55]
*Nicotiana tabacum*	118	118	--	[56]
*P. trichocarpa*	109	62	16	[17]
*S. lycopersicum*	104	54	--	[18]
*V. vinifera*	98	65	49	[57]
*Capsicum annuum*	79	37	41	[51]
*S* *. tuberosum*	--	79	54	[58]
*B. oleracea*	--	37	--	[33]
*B. vulgaris*	104	35	32	

## Data Availability

The data presented in this study are available within the article.

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
