# Peer review of "Genome-Wide Analysis of Q-Type C2H2 ZFP Genes in Response to Biotic and Abiotic Stresses in Sugar Beet"

_biology, 2023, doi:10.3390/biology12101309_

Round 1
Reviewer 1 Report
My comments are in the attached file.

Moderate editing of English language required
Author Response
Dear Reviewer:
We have advised the manuscript according to comments.
The cover letter and response to reviewer have uploaded as a Word file. Please see the attachment.

Reviewer 2 Report
The study focused on the Genome-wide analysis of Q-type C2H2 ZFP genes in response to biotic and abiotic stresses in sugar beet. The study will enhance our understanding of Q-type C2H2 gene family and provide valuable information for biological breading of sugar beet against abiotic and biotic stresses in future. However, there are few limitations which must be addressed.
The abstract should provide brief methods it is only focused on background and results.
Change the spelling “breading” to breeding
In introduction provide the reason that why sugar beet was selected to study the Q-type C2H2 ZFP genes.
What main biotic and abiotic stress challenges are common in sugar beet.
In paragraph 3 of the introduction specify that which kind of biotic and abiotic stress challenges can be control through Q-type C2H2 ZFP.
“important regulatory roles in various biologic processes” specify the processes.
First paragraph of the study lack references and should be cited with the following studies. doi: 10.1038/s41467-022-32364-3, https://doi.org/10.3390/ijms22179175
Line 94-96 provide references.
In last paragraph of the introduction provide aims and objectives of the study based on the study hypothesis, not results.
Provide the reasons “by beet necrotic yellow vein virus (BNYVV) stands one” is it having any connection with abiotic stress?
In some places Arabidopsis is written normal and some in italic. Please be consistent.
Section 2.4 “Analysis of Members of the C2H2 ZFP Gene Family” provide complete details. See the following literature and could be cited here. https://doi.org/10.1016/j.plaphy.2021.01.042
Provide the link “from NCBI SRA database (accession number: PRJNA666117)”
The study concluding that “BvZFPs are involved in the response to abiotic and biotic stresses” these should be specifying in the abstract.
Conclusion must add future perspective of the study.
Check spelling mistakes throughout the MS
Author Response

(The authors gave the same response as above.)

Round 2
Reviewer 2 Report
NA
Fine